# CROSSLMM: DECOUPLING LONG VIDEO SEQUENCES FROM LMMS VIA DUAL CROSS-ATTENTION MECHANISMS

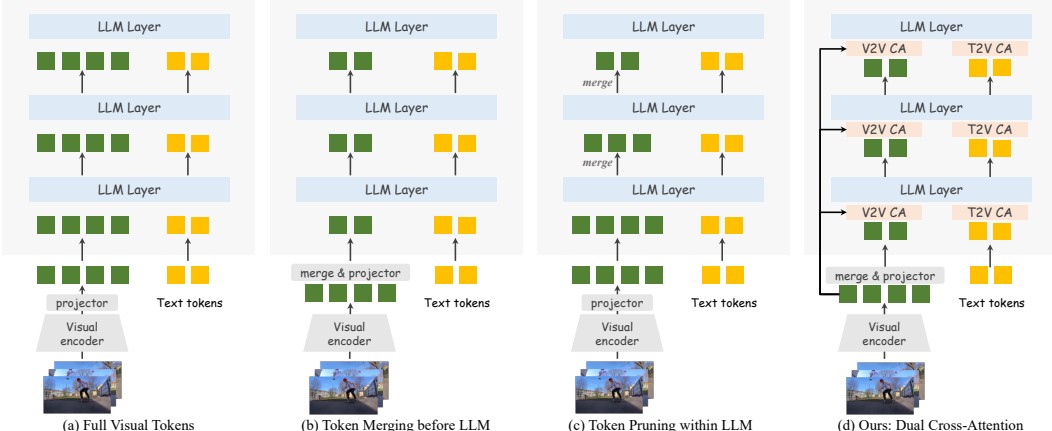

Figure 1: **Comparisons of Different Visual Token Compression Methods.** (a) keeps all visual tokens. (b) and (c) merge visual tokens before and within LLMs, respectively. (d) Our method decouples visual tokens from LLMs with a dual cross-attention mechanism. We first merge visual tokens before LLMs to reduce computational cost in LLMs. Then we propose a Visual-to-Visual Cross-Attention (V2V CA) to preserve fine-grained details of original visual tokens into merged tokens, and a Text-to-Visual Cross-Attention (T2V CA) to enhance text tokens with visual information.

## ABSTRACT

The advent of Large Multimodal Models (LMMs) has significantly enhanced Large Language Models (LLMs) to process and interpret diverse data modalities (e.g., image and video). However, as input complexity increases, particularly with long video sequences, the number of required tokens has grown significantly, leading to quadratically computational costs. This has made the efficient compression of video tokens in LMMs, while maintaining performance integrity, a pressing research challenge. In this paper, we introduce CrossLMM, decoupling long video sequences from LMMs via a dual cross-attention mechanism, which substantially reduces visual token quantity with minimal performance degradation. Specifically, we first implement a significant token reduction from pretrained visual encoders through a pooling methodology. Then, within LLM layers, we employ a visual-to-visual cross-attention mechanism, wherein the pooled visual tokens function as queries against the original visual token set. This module enables more efficient token utilization while retaining fine-grained informational fidelity. In addition, we introduce a text-to-visual cross-attention mechanism, for which the text tokens are enhanced through interaction with the original visual tokens, enriching the visual comprehension of the text tokens. Comprehensive empirical evaluation demonstrates that our approach achieves comparable or superior performance across diverse video-based LMM benchmarks, despite utilizing substantially fewer computational resources.

# 1 INTRODUCTION

Large Multimodal Models (LMMs) (OpenAI, 2024; 2023; Reid et al., 2024; Li et al., 2024e;a; Ma et al., 2025) enhance Large Language Models (LLMs) (Achiam et al., 2023; Touvron et al., 2023; Yang et al., 2024a; Grattafiori et al., 2024) with visual perception capabilities, demonstrating remarkable proficiency in image-language (Jiang et al., 2025; 2024; Zhang et al., 2024d; Fu et al., 2024a) and video-language (Zhou et al., 2024; Fu et al., 2024b; Wu et al., 2024; Patraucean et al., 2024; Hong et al., 2025) tasks. Contemporary research on LMMs predominantly employs an intermediate module, i.e., the projector, to map visual token representations into LLM embedding spaces, which subsequently serve as prefix content for textual tokens into the LLMs as demonstrated in Figure 1 (a). Nevertheless, the increasing complexity of input modalities, particularly long video sequences, generates substantial quantities of visual tokens. This proliferation of tokens necessitates significant computational resources, thereby constraining the applicability of such models in resource-limited environments and long-context scenarios. Consequently, the development of efficient visual compression methods that minimize performance degradation has emerged as a critical research imperative, garnering substantial attention from both academic and industrial communities.

In recent literature, a diverse array of methodologies for visual token compression has emerged, which can be categorized into two principal paradigms as follows. *1) Token Merging before LLM* (Zhong et al., 2024; Huang et al., 2024; Song et al., 2024a), as illustrated in Figure 1 (b), predominantly entails the compression of visual tokens through semantic similarity metrics before feeding into LLMs. Nevertheless, this methodology exhibits significant limitations: it fails to adequately identify visual tokens corresponding to relevant textual elements, compromising spatial relationships and attenuating the model's capacity for comprehensive image interpretation. *2) Token Pruning within LLM* (Shang et al., 2024; Ye et al., 2024; Zhao et al., 2024; Sun et al., 2025), as depicted in Figure 1 (c), implements a systematic reduction of visual tokens within the input layers of LLMs. This methodology typically employs the quantification of attention weights between visual and textual tokens, subsequently eliminating those tokens that manifest lower weight values. While this approach effectively illuminates multi-modal interactions, it might instead overlook crucial semantic information within the visual modality. Furthermore, both methods experience significant performance degradation as the number of visual tokens decreases. Consequently, an intriguing question arises: *Is it possible to maintain comparable performance while significantly reducing the number of video tokens?*

To achieve this objective, we introduce an efficient architecture designated as **CrossLMM**. The fundamental component comprises a dual cross-attention module. In the visual domain, we initially compress image tokens derived from the visual encoder along the spatial dimension to reduce the token quantity inputted into LLMs. To facilitate comprehensive image information capture, we implement a visual-to-visual (V2V) cross-attention mechanism within the LLM layer. This process enables sufficient interaction between the compressed visual tokens (as queries) and the original long-sequence visual representations (as keys and values). Correspondingly, in the textual domain, we employ a text-to-visual (T2V) cross-attention mechanism to enhance text tokens (as queries) with multimodal information from original visual tokens (as keys and values). This further complements the text generation process with fine-grained visual semantics. Through the implementation of this dual cross-attention mechanism, we endeavor to maintain the fidelity of the original visual tokens, effectively mitigating performance deterioration, while simultaneously achieving substantial token compression.

We evaluate CrossLMM on an extensive spectrum of video-based multimodal benchmarks. Our model, utilizing very few tokens (1 or 9 or 16), demonstrates promising performance across various video assessments, including VideoMME (Fu et al., 2024b) and MLVU (Zhou et al., 2024) and so on. These findings substantiate that the implementation of our dual cross-attention module, coupled with a restricted number of visual tokens, enables LMMs to effectively address diverse visual tasks.

Our contributions can be summarized as follows:

1. We introduce CrossLMM, which effectively compacts long video sequences into efficient representations via a dual cross-attention mechanism.

2. We propose the V2V and T2V cross-attention layers, providing semantics from original visual tokens for compact visual tokens and text tokens, respectively.

3. CrossLMM archives comparable or state-of-the-art performance across various video understanding benchmarks with only few visual tokens, demonstrating superior efficiency.

## 2 RELATED WORKS

### 2.1 LARGE MULTIMODAL MODELS

Large language models (LLMs) (Achiam et al., 2023; Touvron et al., 2023; Yang et al., 2024a; Grattafiori et al., 2024) trained on extensive datasets have demonstrated remarkable capabilities in text understanding and generation tasks, establishing the foundation for developing multi-modal LLMs. Among large multimodal models (LMMs) (OpenAI, 2024; 2023; Reid et al., 2024; Li et al., 2024e;a; Ma et al., 2025; 2024; An et al., 2024; Lin et al., 2025), LLaVA (Liu et al., 2023) has emerged as the predominant architecture, valued for its data efficiency and streamlined design. This approach incorporates a projector that effectively bridges visual and language modalities, while employing instruction-tuning for both the projector and the LLM using comprehensive instruction-following datasets. Recent scholarly work (Li et al., 2024a; Chen et al., 2024c;b; Li et al., 2024b) has sought to enhance model performance by implementing high-resolution visual inputs. Concurrently, video-based LLMs (Zhang et al., 2024e;f; Liu et al., 2024; Xue et al., 2024; Shen et al., 2024) have advanced through the extension of visual instruction tuning datasets to accommodate video modality. However, the substantial number of tokens generated by high-resolution and video inputs presents significant computational challenges. Consequently, there is a pressing need to develop efficient methods for visual token compression to address these limitations.

### 2.2 VISUAL TOKEN COMPRESSION

With high-resolution image and video inputs, the token count has increased substantially, often exceeding textual token quantities by one to two orders of magnitude. Consequently, the efficient compression of visual tokens has emerged as a critical research challenge. Prior approaches (Zhong et al., 2024; Huang et al., 2024; Song et al., 2024a; Shang et al., 2024; Ye et al., 2024; Zhao et al., 2024; Sun et al., 2025) addressing this issue can be categorized into two principal directions. The first category encompasses methods that compress tokens generated by visual encoders or implement efficient projections of visual modalities. For instance, LLaMA-VID (Li et al., 2024e) employs a QFormer (Li et al., 2023) architecture to compress visual tokens into a minimal representation of two tokens. Similarly, Deco (Yao et al., 2024) implements adaptive pooling at the image patch level. However, these approaches lack integration with textual information for guided compression. Alternative methods gradually reduce visual tokens within Large Language Models (LLMs), yet such approaches frequently compromise the semantic integrity of visual information. Therefore, we propose a dual cross-attention module that simultaneously updates text-relevant visual tokens and visually-relevant textual tokens, thereby effectively compressing visual representations while minimizing information loss.

## 3 METHOD

In this section, we illustrate the details of our CrossLMM for multi-modal video understanding. We first describe the overall pipeline in Section 3.1. Then, in Section 3.2 and Section 3.3, we respectively elaborate on the proposed designs of visual-to-visual and text-to-visual cross-attention mechanisms.

### 3.1 OVERALL ARCHITECTURE

Our proposed approach, CrossLMM, inspired by current mainstream architecture LLaVA (Liu et al., 2023), constructs around three core components, including the frame-wise visual encoder, the visual-language projector, and the Large Language Model (LLM), as demonstrated in Figure 2.

**Visual Feature Encoding.** For visual feature extraction, we employ an image-based encoder rather than video-based architectures, such as VideoMAE (Tong et al., 2022) or Video-Swin (Liu et al., 2022). This approach extracts features from individual frames sequentially. Specifically, we utilize the pre-trained SigLIP2 (Tschannen et al., 2025) vision encoder for encoding visual information.

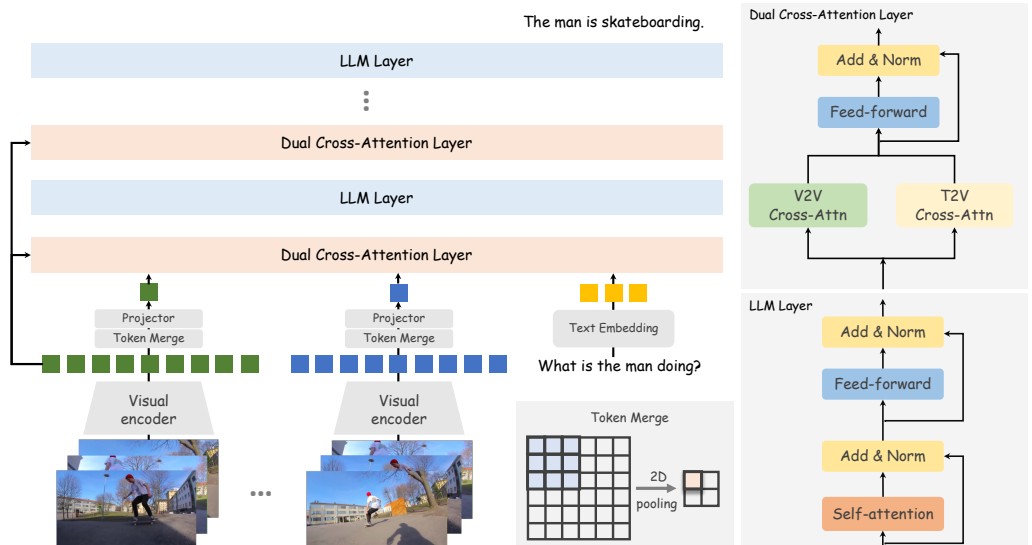

Figure 2: **Architecture of CrossLMM,** which consists of a visual encoder, a visual projector and an LLM. For a pretrained LLM, we insert the proposed Dual Cross-Attention Layer (DCAL) to it every $n$ layers. The DCAL is a variant of general cross-attention layer with two parallel blocks: Visual-to-Visual (V2V) Cross-Attention and Text-to-Visual (T2V) Cross-Attention. Both V2V Cross-Attn and T2V Cross-Attn aggregate fine-grained information from the original visual tokens to produce visual-enhanced video tokens and text tokens.

Our methodology is justified by two key considerations: 1) image encoders normally contain better generalization capabilities with larger-scale training, and 2) the temporal information can be retained by concatenating multiple frame representations. Given an input video-text pair, we sample $T$ frames $v \in \mathbb{R}^{T \times 3 \times H \times W}$ from the video clip and apply the visual encoder to extract image features. This process can be formulated as:

$$\mathcal{V} = \{x_i = \mathcal{F}(v_i) \mid \forall i = 1, \ldots, T\} \tag{1}$$

Here, $\mathcal{V} \in \mathbb{R}^{T \times N \times D}$ corresponds to the feature representations of the $T$ input frames. The visual encoder $\mathcal{F} : \mathbb{R}^{3 \times H \times W} \to \mathbb{R}^{N \times D}$ processes each RGB frame $v_i$ (with spatial resolution $H \times W$) to generate $x_i \in \mathbb{R}^{N \times D}$, where $N$ denotes the cardinality of visual tokens and $D$ specifies the latent embedding dimension per token.

**Initial Token Merge.** After that, the bilinear pooling operator $\mathcal{B} : \mathbb{R}^{N \times D} \to \mathbb{R}^{N/9 \times D}$ is applied preceding the projection operation, performing $3 \times 3$ local patch aggregation. This spatially downsamples the token grid from original $27 \times 27$ ($N = 729$) to $3 \times 3$ ($N = 9$) resolution through non-overlapping fusion of spatially adjacent patches, while preserving the feature dimension $D$ through parameter-free structural compression. This architecture introduces geometrically meaningful inductive bias through structural token aggregation while maintaining the integrity of inter-patch spatial relations, which also enhances computational efficiency.

**Visual-language Projector.** The cross-modal projection module $\Phi : \mathbb{R}^D \to \mathbb{R}^{D'}$ is architected as a parameterized two-layer MLP (MLP : $\mathbb{R}^D \to \mathbb{R}^{d_h} \to \mathbb{R}^{D'}$), operating on visual tokens $x_i \in \mathbb{R}^D$ through successive nonlinear transformations (GeLU (Hendrycks & Gimpel, 2016) activation with layer normalization). This learnable transformation achieves bidirectional semantic alignment by establishing: (1) an injective mapping from vision feature space $\mathcal{V}$ to the LLM's textual embedding manifold $\mathcal{T}$, and (2) a differentiable inverse approximation $\Phi_{\text{approx}}^{-1}$ preserving topological consistency. As the pivotal interface enabling visio-linguistic fusion, $\Phi$ induces latent space isomorphism through $\ell_2$-sphere projection regularization, thus constituting the mathematical foundation for constructing joint embedding space.

**Large Language Model.** The architecture builds upon a standard decoder-only LLM foundation, augmented with a hierarchical cross-modal integration scheme. At every $K$ decoder layers, we introduce dual-stream attention mechanisms:

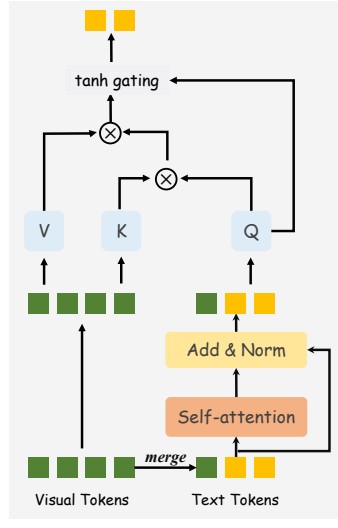

(a) V2V Cross-Attention    (b) T2V Cross-Attention

Figure 3: **Implementation Details of Dual Cross-Attention.** For a detailed illustration, please refer to Sec. 3.2 and Sec. 3.3.

- **Visual-to-Visual**: Positioned after textual self-attention, this module computes

$$\text{Attn}(Q_{v_p}, K_{v_o}, V_{v_o}) = \text{Softmax}\left(\frac{Q_{v_p} K_{v_o}^{\top}}{\sqrt{d_k}}\right) V_{v_o} \tag{2}$$

where $Q_{v_p}$ derives from efficient visual representations and $\{K_{v_o}, V_{v_o}\}$ are projected original visual tokens. This enables text-conditioned vision-guided fine-grained refinement of efficient visual representations from original visual tokens.

- **Text-to-Visual**: Operating in parallel, it establishes

$$\text{Attn}(Q_{\text{text}}, K_{v_o}, V_{v_o}) \tag{3}$$

creating visual-conditioned text feature updating.

This staggered integration strategy implements multi-round cross-modal fusion through alternating attention directions, where $\mathcal{V}$-Cross-Attn extracts text-related visual specifics to ground language generation, while $\mathcal{T}$-Cross-Attn maintains semantic coherence through visual-related linguistic feedback forming a coupled attention dynamical system. Further details of this $\mathcal{V}$-Cross-Attn and $\mathcal{T}$-Cross-Attn are provided in Section 3.2 and Section 3.3, respectively.

## 3.2 Visual-to-Visual Cross-attention

To boost both the multi-modal and multi-frame feature fusion, we introduce visual-to-visual (V2V) cross attention module for text-condictioned visual aggregation from coarse-grained to fine-grained. For a multimodal vision-language model, the projected and pooled visual feature $\mathcal{V} \in \mathbb{R}^{L_v \times D'}$ would concat with text feature $\mathcal{T} \in \mathbb{R}^{L_t \times D'}$ into the LLM, where $D'$ denotes the embedding dimension of LLM, and $L_v, L_t$ represents the length of visual and text tokens, respectively. As shown in Figure 3(a), the V2V module take the visual feature $\mathcal{V}$ (pooled and projected), and text features $\mathcal{T}$ as input $\mathcal{I} = [\mathcal{V}; \mathcal{T}]$, where $[;]$ denotes concatenation. we adopt gated cross-modal fusion mechanisms for input tokens to progressively capture multi-modal information. There consists of two stages, self-attention processing and cross-attention fusion. That can be formulated as,

$$\mathcal{I}' = \text{LayerNorm}(\text{SelfAttn}(\mathcal{I}) + \mathcal{I}), \tag{4}$$

where $\mathcal{I}'$ represents the integrated representation of visual and textual features. Subsequently, we extract the coarse text-conditioned visual feature $\mathcal{I}'[: L_v]$. To obtain a more fine-grained visual representation, we employ $\mathcal{I}'[: L_v]$ as query tokens, as expressed by:

$$Q = W_q \mathcal{I}'[: L_v] \in \mathbb{R}^{L_v \times D'}, \tag{5}$$

while utilizing the original tokens as the Key and Value tokens:

$$K, V = (W_k / W_v)\mathcal{V} \in \mathbb{R}^{T \times N \times D'}, \tag{6}$$

where $W_q$, $W_k$, and $W_v$ denote linear projection matrices. The fine-grained visual feature is then acquired through cross-attention and gating mechanisms, which can be formulated as:

$$\mathcal{I}''[: L_v] = \text{Attn}(Q, K, V) + \gamma * \mathcal{I}'[: L_v] \tag{7}$$

where $\gamma \in [-1, 1]$ is a learnable gating parameter. Through this process, we derive a hierarchical text-conditioned visual representation that systematically progresses from coarse-grained to fine-grained feature abstraction.

### 3.3 TEXT-TO-VISUAL CROSS-ATTENTION

To enhance the visual comprehension of the text tokens, we implement a text-to-visual (T2V) cross-attention mechanism that facilitates interaction between text tokens and original visual tokens. While structurally parallel to the V2V module previously described, the T2V module operates in the complementary direction, transforming text token representations through visual context. The mathematical formulation follows a similar pattern:

$$Q = W_q \mathcal{I}'[L_v : L_v + L_t] \in \mathbb{R}^{L_t \times D'}, \tag{8}$$

where text tokens serve as queries that attend to the visual information. The original visual tokens function as Key and Value tokens:

$$K, V = (W_k / W_v)\mathcal{V} \in \mathbb{R}^{T \times N \times D'}, \tag{9}$$

with $W_q$, $W_k$, and $W_v$ representing the respective linear projection matrices. The enhanced text features are computed through cross-attention and modulated by a learnable gating mechanism:

$$\mathcal{I}'[L_v : L_v + L_t] = \text{Attn}(Q, K, V) + \gamma * \mathcal{I}'[L_v : L_v + L_t], \tag{10}$$

where $\gamma \in [-1, 1]$ controls the influence of the visually-contextualized information. This mechanism complements the V2V module by providing the reciprocal flow of information, resulting in visual-conditioned text representations that capture multi-level semantic relationships between the two modalities.

## 4 EXPERIMENTS

### 4.1 EXPERIMENT SETTINGS

**Implementation Details.** CrossLMM employs SigLIP2 (Tschannen et al., 2025) as the vision encoder. Qwen2.5-1.5B-Instruct (Yang et al., 2024a) serves as the large language model (LLM) for CrossLMM-2B, while Qwen2.5-7B-Instruct (Yang et al., 2024a) is utilized for CrossLMM-7B. For the visual-to-visual module, we integrate connections at every $K$ layers within the LLM, following the same approach as the text-to-visual module. During the pretraining phase, we implement a learning rate of $1 \times 10^{-4}$, while in the instruction tuning stage, the learning rate is adjusted to $1 \times 10^{-5}$, with the exception of the Vision Transformer (ViT) component, which uses $2 \times 10^{-6}$. Our model undergoes training only one epoch during both pretraining and instrcution tuning stage. More details can be found in supplementary materails.

**Evaluation Benchmarks.** To validate CrossLMM's general capabilities, we evaluate our model on five video understanding benchmarks spanning short video benchmarks, long video benchmarks, and comprehensive benchmarks. These include two short video benchmarks: MVBench (Li et al., 2024c) and Perception Test (Patraucean et al., 2024), and two long video benchmarks: LongVideoBench (Wu et al., 2024) and MLVU (Zhou et al., 2024), and a comprehensive benchmark, VideoMME (Fu et al., 2024b), covering videos ranging from minute-level to hour-level.

Table 1: **Comprehensive evaluation of video understanding models.** Performance comparison across multiple video understanding benchmarks for different categories of models (proprietary, small-size LMMs, general open-source LMMs, and specialized long video LMMs). The CrossLMM model (highlighted) achieves competitive or superior performance while using significantly fewer tokens per frame (1 or 9 or 16). † represents the vision encoder is ViT-G.

| Model | Size | #tokens per frame | MVBench Avg | PerceptionTest Val | LongVideoBench Val | MLVU M-Avg | VideoMME w/o sub. | VideoMME w sub. |
|---|---|---|---|---|---|---|---|---|
| Avg. Duration | | | 16s | 23s | 473s | 651s | 1010s | 1010s |
| *Proprietary Models* | | | | | | | | |
| GPT4-V (OpenAI, 2023) | - | - | 43.7 | - | 59.1 | 49.2 | 59.9 | 63.3 |
| GPT4-o (OpenAI, 2024) | - | - | 64.6 | - | 66.7 | 64.6 | 71.9 | 77.2 |
| Gemini-1.5-Pro (Reid et al., 2024) | - | - | 60.5 | - | 64.0 | - | 75.0 | 81.3 |
| *Small Size LMMs* | | | | | | | | |
| Qwen2-VL (Wang et al., 2024a) | 2B | - | 63.2 | - | - | - | 55.6 | 60.4 |
| InternVL2.5 (Chen et al., 2024b) | 2B | 256 | **68.8** | - | 46.0 | **61.4** | 51.9 | 54.1 |
| **CrossLMM** | 2B | 1 | 58.6 | 58.1 | 47.3 | 57.6 | 55.5 | 58.9 |
| **CrossLMM** | 2B | 9 | 63.5 | **63.5** | **51.8** | 61.0 | **59.1** | **61.3** |
| *Open-Source LMMs* | | | | | | | | |
| VideoLLaMA2 (Cheng et al., 2024) | 7B | 72 | 54.6 | 51.4 | - | 48.5 | 47.9 | 50.3 |
| VideoLLaMA2 (Cheng et al., 2024) | 72B | 72 | 62.0 57.5 | - | - | 62.4 | 64.7 | |
| VideoChat2-HD (Li et al., 2024c) | 7B | 72 | 62.3 | - | - | 47.9 | 45.3 | 55.7 |
| InternVideo2-HD (Wang et al., 2024c) | 7B | 72 | 67.2 | 63.4 | - | - | 49.4 | - |
| IXComposer-2.5 (Zhang et al., 2024a) | 7B | 400 | 69.1 | 34.4 | - | 37.3 | 55.8 | 58.8 |
| InternVL2 (Chen et al., 2024c) | 8B | 256 | 65.8 | - | 54.6 | 64.0 | 54.0 | 56.9 |
| InternVL2 (Chen et al., 2024c) | 76B | 256 | 69.6 | - | 61.1 | 69.9 | 61.2 | 62.8 |
| InternVL2.5 (Chen et al., 2024b) | 8B | 256 | 72.0 | - | 60.0 | 68.9 | 64.2 | 66.9 |
| Qwen2-VL (Wang et al., 2024a) | 7B | - | 67.0 | 62.3 | - | - | 63.3 | 69.0 |
| Qwen2-VL (Wang et al., 2024a) | 72B | - | 73.6 | 68.0 | - | - | 71.2 | 77.8 |
| LLaVA-NeXT-Video (Zhang et al., 2024e) | 7B | 144 | 53.1 | 48.8 | 49.1 | - | - | 46.5 |
| LLaVA-OneVision (Li et al., 2024a) | 7B | 196 | 56.7 | 57.1 | 56.3 | 64.7 | 58.2 | 61.5 |
| LLaVA-OneVision (Li et al., 2024a) | 72B | 196 | 59.4 | 66.9 | 61.3 | 68.0 | 66.2 | 69.5 |
| LLaVA-Video (Zhang et al., 2024f) | 7B | 676 | 58.6 | 67.9 | 58.2 | 70.8 | 63.3 | 69.7 |
| *Open-Source Long Video LMMs* | | | | | | | | |
| LLaMA-VID (Li et al., 2024e) | 7B | 2 | 41.9 | 44.6 | - | 33.2 | 25.9 | - |
| Kangaroo Liu et al. (2024) | 8B | 256 | 61.0 | - | 54.8 | 61.0 | 56.0 | 57.6 |
| LongVILA (Xue et al., 2024) | 7B | 196 | 67.1 | 58.1 | 57.1 | - | 60.1 | 65.6 |
| LongVA (Zhang et al., 2024b) | 7B | 144 | - | - | - | 56.3 | 52.6 | 54.3 |
| LongLLaVA (Wang et al., 2024b) | 9B | 144 | 49.1 | - | - | - | 43.7 | - |
| LongVU (Shen et al., 2024) | 7B | 64 | 66.9 | - | - | 65.4 | - | 60.6 |
| **CrossLMM** | 7B | 1 | 62.7 | 64.8 | 54.5 | 64.0 | 61.3 | 63.7 |
| **CrossLMM** | 7B | 9 | 68.2 | 68.4 | 56.0 | 67.2 | 62.6 | 64.7 |
| **CrossLMM †** | 7B | 16 | **68.8** | **71.4** | **60.4** | **70.7** | **65.4** | **67.6** |

## 4.2 COMPARISON WITH STATE-OF-THE-ART METHODS

In this section, we present a comparative analysis of CrossLMM against state-of-the-art LMMs, encompassing commercial implementations (OpenAI, 2023; 2024; Reid et al., 2024), open-source LMMs (Cheng et al., 2024; Li et al., 2024c; Wang et al., 2024c; Zhang et al., 2024a; Chen et al., 2024c;b; Wang et al., 2024a; Zhang et al., 2024e; Li et al., 2024a; Zhang et al., 2024f), and specialized open-source long video LMMs (Li et al., 2024e; Liu et al., 2024; Xue et al., 2024; Zhang et al., 2024b; Wang et al., 2024b; Shen et al., 2024).

Our experimental results, as presented in Table 1, demonstrate several significant findings in the landscape of video understanding models. CrossLMM exhibits remarkable efficiency-performance trade-offs compared to existing models. Despite utilizing only 9 tokens per frame—significantly fewer than competitors that employ between 64 and 676 tokens—CrossLMM achieves competitive performance across multiple benchmarks. Moreover, CrossLMM with 7B parameters and 16 tokens per frame surpasses several mainstream open-source LMMs, while maintaining lower memory and computation overhead. This efficiency is consistently observed under both 2B and 7B model scales, which highlight the potential of CrossLLM that designed efficiently that balance token efficiency and performance for practical video understanding applications.

## 4.3 EFFICIENCY ANALYSIS

Figure 4 presents a comprehensive efficiency comparison between LLaVA-OV (Li et al., 2024a) and CrossLMM across different frame numbers (32, 64, 128, and 256) with 8 H800. The analysis focuses on three critical metrics: CUDA memory consumption, computational complexity, and prefill time.

**Memory Efficiency.** As shown in Figure 4(a), CrossLMM demonstrates remarkable memory efficiency compared to LLaVA-OV. At 32 frames, CrossLMM requires only 1,753MB of CUDA memory, which is 63.9% less than the 4,858MB needed by LLaVA-OV. This memory advantage becomes increasingly pronounced as the frame count increases. At 256 frames, CrossLMM consumes just

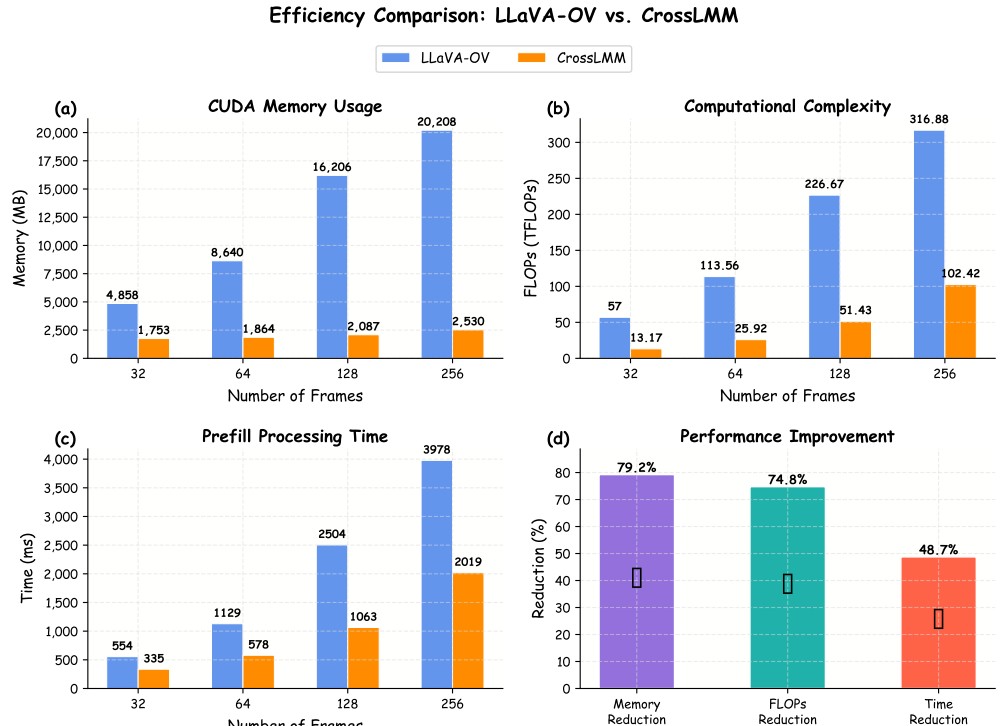

Figure 4: **Efficiency: CrossLMM vs. LLaVA-OV.** CrossLMM significantly outperforms LLaVA-OV in memory, computation, and speed, especially as frame count increases.

2,531MB, representing an 87.5% reduction from LLaVA-OV's 20,208MB. Notably, the memory consumption of CrossLMM scales much more gradually with increasing frame counts, exhibiting an almost sub-linear growth pattern compared to LLaVA-OV's near-linear growth.

**Computational Efficiency.** Figure 4(b) illustrates the computational requirements measured in TFLOPs. CrossLMM consistently achieves lower computational complexity across all frame counts. At 32 frames, CrossLMM requires 13.17 TFLOPs, which is 76.9% less than LLaVA-OV's 57 TFLOPs. The computational advantage persists at higher frame counts, with CrossLMM requiring 102.42 TFLOPs at 256 frames compared to LLaVA-OV's 316.88 TFLOPs, representing a 67.7% reduction. This substantial decrease in computational demands contributes to CrossLMM's overall efficiency and potentially enables deployment on resources-limited devices.

**Processing Time Efficiency.** The prefill processing time, depicted in Figure 4(c), shows that CrossLMM consistently outperforms LLaVA-OV (Li et al., 2024a) in terms of speed. At 32 frames, CrossLMM completes processing in 335ms, which is 39.6% faster than LLaVA-OV's 554.91ms. This advantage is maintained across higher frame counts, with CrossLMM processing 256 frames in 1,975ms compared to LLaVA-OV's 3,978.83ms, representing a 50.4% reduction in processing time. The time efficiency of CrossLMM is particularly significant for real-time applications where latency is a critical factor.

**Overall Efficiency Improvement.** Figure 4(d) summarizes the average performance improvements achieved by CrossLMM across all tested frame counts. The most substantial gain is observed in memory usage, with an average reduction of 79.2%, followed by computational requirements at 74.8%, and processing time at 48.7%. These consistent improvements indicate that CrossLMM's architectural design effectively addresses the efficiency limitations of previous approaches.

The efficiency gains can be attributed to CrossLMM's novel approach to multi-modal processing, which employs a strategic token pruning to reduce redundancy in the video frame representations. On top of this, by leveraging V2V and T2V modules capture the multi-modal representations efficiently, CrossLMM minimizes computational overhead while maintaining model performance. These significant efficiency improvements enable CrossLMM to process longer video sequences with substantially lower resource requirements, making it more suitable for practical applications and deployment in resource-constrained environments.

Table 2: **Ablation studies on key components of our method.** (a) Visual-to-Visual (V2V) and Text-to-Visual (T2V) modules (✓: applied, ✗: not applied). (b) T2V and V2V modules insertion frequency $K$. (c) T2V activation stage (PT: pretraining, SFT: instruction-tuning).

(a) V2V and T2V modules

| Modules | | VideoMME | MVBench |
|---|---|---|---|
| V2V | T2V | Overall | Avg. |
| ✗ | ✗ | 47.2 | 47.5 |
| ✗ | ✓ | 47.3 | 47.7 |
| ✓ | ✗ | 48.4 | 48.9 |
| ✓ | ✓ | **48.7** | **49.9** |

(b) Insertion frequency $K$

| $K$ | VideoMME | MVBench |
|---|---|---|
| | Overall | Avg. |
| None | 47.2 | 47.5 |
| 1 | 48.5 | 47.8 |
| 2 | 48.5 | 49.7 |
| 4 | **48.7** | **49.9** |
| 8 | 47.7 | 48.5 |

(c) T2V activation stage

| Modules | | VideoMME | MVBench |
|---|---|---|---|
| PT | SFT | Overall | Avg. |
| ✗ | ✗ | 48.4 | 48.9 |
| ✓ | ✓ | 48.0 | 47.8 |
| ✗ | ✓ | **48.7** | **49.9** |

## 4.4 ABLATION STUDY

In this section, we present comprehensive ablation experiments to evaluate the contributions of individual components. Given computational resource constraints, we employ the CrossLMM-2B model as the foundation with 1 tokens for our ablation studies. During the pretraining phase, we utilize a balanced corpus comprising 3.75 million image-text pairs and 1.25 million video-text pairs, maintaining a consistent sampling ratio of 1:1 between modalities. For the instruction-tuning phase, we leverage the established LLaVA-Video 178K dataset to optimize model performance.

**Visual-to-Visual and Text-to-Visual:** The V2V module is specifically designed to capture text-conditioned fine-grained visual features from the original visual tokens. On top of this, the T2V module is designed to enhance the visual comprehension of the text information. As shown in Table 2a, in the absence of the V2V or T2V modules, performance decreases across all benchmarks, directly demonstrating the critical roles of the V2V and T2V modules in extracting fine-grained visual information and text features.

**Insertion frequency:** We conducted systematic ablation experiments by varying the insertion frequency $K$, which is shwon in Table 2b. When $K = 1$ (indicating maximum insertion frequency), we observed performance significantly below our proposed configuration, primarily attributable to the excessive number of parameters introduced, which compromises the inherent capabilities of the original LLM. Similarly, when $K = 8$, performance remains suboptimal compared to our method, suggesting that at this insertion frequency, the fine-grained visual information becomes overly sparse.

**T2V activation stage:** Our experimental design incorporates three configurations of the Text-to-Visual (T2V) module: (1) omission of the T2V module during both pretraining and instruction-tuning phases; (2) implementation of the T2V module throughout both pretraining and instruction-tuning phases; and (3) inclusion of the T2V module during pretraining but exclusion during instruction-tuning. The ablation study results, presented in Table 2c, yield two significant findings. First, the absence of the T2V module in both stages results in diminished performance across all benchmarks compared to our optimal configuration, demonstrating the module's efficacy in enhancing the visual representation of text tokens. Second, while incorporating the T2V module solely in the pretraining stage produces improvements, the efficacy is significantly attenuated due to the considerable textual noise present in pretraining data, such as alt-text descriptions.

## 5 CONCLUSION

We present CrossLMM, which offers a rigorous solution to the computational inefficiency that has long constrained LMMs when processing extended video sequences. Our CrossLMM framework demonstrates that through judicious token reduction strategies and architectural innovations, significant computational efficiency can be achieved while preserving model fidelity. The dual cross-attention mechanism, comprising V2V cross attention for maintaining fine-grained representational integrity and T2V cross attention for enhanced multimodal comprehension, constitutes a methodological advancement in token management for the practical real-world multimodal systems.

## REPRODUCIBILITY

To ensure the reproducibility of our research, we provide a comprehensive description of our methodology. The experimental setup, including the benchmarks, model configurations, and hyper-parameters, is detailed in the Experiments section. Furthermore, the Appendix specifies the datasets and strategies employed for training. The source code and models will be made publicly available upon publication.

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

## A STATEMENT ON THE USE OF AI TOOLS

Large Language Models (LLMs) were utilized in the preparation of this manuscript. Their role was strictly limited to that of a writing assistant, aiding in grammar correction, sentence refinement, and improvement of overall readability.

## B IMPLEMENTATION DETAILS

### B.1 TRAINING STRATEGIES AND DATA

**Training Strategies.** In contrast to current mainstream LMMs (Wang et al., 2024a; Chen et al., 2024b;c), which typically employ intricate three- or four-stage training pipelines, we argue that such methodologies introduce unnecessary complexity and accessibility barriers. To address this, we adopt a streamlined two-stage training framework inspired by earlier LMM practices (Liu et al., 2023; Zhu et al., 2023).

- **Stage I: Vision-Language Pretraining.** During this stage, CrossLMM aligns visual and linguistic representations using visual captioning datasets. Notably, due to the inherent noise in textual data, the text-to-visual module is intentionally excluded at this phase. Training instead focuses exclusively on optimizing the projection and visual-to-visual modules, while both the vision encoder and the LLM remain frozen to preserve their pretrained knowledge.

- **Stage II: Instruction-tuning.** In this stage, CrossLMM undergoes comprehensive training to execute diverse video-related tasks (e.g., question answering, multiple-choice assessment, etc.) utilizing computationally efficient vision tokens based on instruction-tuning data. The model's parameters are jointly optimized through end-to-end training methodology.

**Training Data.** In contrast to numerous contemporary LMMs that rely extensively on proprietary in-house datasets, CrossLMM exclusively utilizes publicly available open-source data for its development and training.

- **Stage I: Pretraining Data.** The pre-training corpus for CrossLMM consists of two primary components: (1) 15M image-text pairs from CapsFusion (Yu et al., 2024), and (2) 5M video-text pairs sourced from InternVid (Wang et al., 2023). A sampling ratio of 1:1 is implemented between image-text and video-text pairs throughout the training process.

- **Stage II: Instruction Data.** The instruction-tuning data consists of about 3 million samples including image instruction data, short video instruction data and long video instruction data.

    - **Image Instruction data.** In our methodology, we employed a corpus of single-image instruction data derived from multiple established datasets, specifically LLaVA-NeXT (Zhang et al., 2024e), ALLaVA (Chen et al., 2024a), and ShareGPT4-o (Wang et al., 2024c; Chen et al., 2024c). To enhance the model's capacity for processing complex visual inputs, we further augmented our training regime with multi-image sequential data obtained from LLaVA-Interleave (Li et al., 2024b). This comprehensive data integration approach facilitated a more robust visual-linguistic representational framework.

    - **Short Video Instruction data.** For the instruction fine-tuning phase of our research, we predominantly employed short video sequences from VideoChat2 (Li et al., 2024c) and InternVideo2 (Wang et al., 2024c) datasets. To enhance the model's comprehension capabilities, we further supplemented the training corpus with annotations derived via GPT4-o from several established datasets, including ShareGPT4o (Wang et al., 2024c; Chen et al., 2024c), VideoChatGPT-Plus (Maaz et al., 2024), LLaVA-Video-178K (Zhang et al., 2024f), and LLava-Hound (Zhang et al., 2024c). This methodical integration of diverse video-based instructional data facilitated the development of a more comprehensive temporal-visual understanding framework.

    - **Long Video Instruction data.** In our experimental framework, we primarily utilized long-form video instruction datasets, specifically those sourced from MovieChat (Song et al., 2024b) and Vript (Yang et al., 2024b), supplemented by LongVid corpus (Li et al., 2024d). This methodological approach to data selection enabled comprehensive training on extended temporal sequences, thereby facilitating the model's capacity to process,

interpret, and generate responses to complex narrative structures and sustained visual information across prolonged video segments.

## B.2 TRAINING HYPERPARAMETERS.

As delineated in Table 3, we present a comprehensive documentation of the training protocols and associated hyperparametric configurations implemented across the sequential developmental stages of our CrossLMM model.

Table 3: **Training details of each training stage for the CrossLMM-2B model.**

| | | Stage 1 | Stage 2 |
|---|---|---|---|
| *Vision* | **Resolution×Num. frames** | 384 | 384 ×8 |
| | #Tokens | $9 \times 32$ | $9 \times (64{\sim}512)$ |
| *Data* | **Dataset** | Image & Short Video | (Multi)-Image & Short/Long Video |
| | #Samples | 20M | 3M |
| *Model* | **Trainable** | Projector & T2V Cross Attention | Full Model |
| | #Parameters | 43M | 2B |
| *Training* | **Batch Size** | 512 | 128 |
| | **LR** of *vision encoder* | $1\times10^{-3}$ | $2\times10^{-6}$ |
| | **LR** *of connector & LLM* | $1\times10^{-3}$ | $1\times10^{-5}$ |
| | **Epoch** | 1 | 1 |

## C  MORE EXPERIMENTS

### C.1  ABLATION STUDY.

**Pretraining data volume analysis.**   To establish a robust experimental foundation, we utilize the CrossLMM-2B model for conducting comprehensive ablation studies. For the instruction fine-tuning phase, we incorporate the LLaVA-Video-178K dataset. Our investigation focuses on the impact of varying pretraining data volume, with results presented in Table 4. This analysis aims to determine optimal data exposure during the critical pretraining stage.

**Vision encoder trainability investigation.**   Table 5 presents our systematic investigation into vision encoder parameter adaptation strategies. We examine the differential effects between maintaining a frozen encoder versus allowing parameter updates during training. This comparison elucidates the significance of vision encoder plasticity on cross-modal representation learning and downstream performance across multiple evaluation benchmarks.

### C.2  COMPUTATIONAL EFFICIENCY ANALYSIS

The comprehensive efficiency analysis presented in Table 6 demonstrates how the CrossLMM-2B model scales with increasing frame counts under controlled experimental conditions. Memory utilization exhibits a sub-linear growth pattern, increasing from 715.61 MB at 32 frames to 1490.25 MB at 256 frames—approximately a 2.08× increase despite the 8× expansion in input dimensionality. This favorable scaling characteristic indicates efficient memory management within the model architecture.

Computational requirements, quantified in TFLOPs (Trillion Floating Point Operations), demonstrate a near-linear relationship with frame count, escalating from 10.93 TFLOPs to 86.82 TFLOPs across the evaluated range. This represents an 7.94× increase, closely approximating the theoretical linear scaling factor. Similarly, the prefill processing latency exhibits proportional growth, with measured times increasing from 264.06 ms to 1698.12 ms as the frame count expands.

These empirical measurements suggest that CrossLMM-2B maintains computational efficiency across varying temporal resolutions, with memory utilization scaling particularly well. Such char-

Table 4: **Quantitative analysis of pretraining data volume impact on multi-benchmark performance.**

| Data Volume | MVBench | PerceptionTest | LongVideoBench | VideoMME | VideoMME |
|---|---|---|---|---|---|
| | *Avg* | *Val* | *Val* | *w/o sub.* | *w sub.* |
| 10M samples | 51.4 | 60.4 | 48.1 | 53.1 | 56.4 |
| 15M samples | 51.3 | 60.5 | 48.5 | 52.7 | 56.5 |
| 20M samples | **52.1** | **60.8** | **49.2** | **54.2** | **57.0** |

Table 5: **Comparative analysis of vision encoder training strategies.**

| Parameter State | MVBench | PerceptionTest | LongVideoBench | VideoMME | VideoMME |
|---|---|---|---|---|---|
| | *Avg* | *Val* | *Val* | *w/o sub.* | *w sub.* |
| Frozen | 60.8 | 61.3 | 51.0 | 56.7 | 59.2 |
| Trainable | **63.5** | **63.5** | **51.8** | **59.1** | **61.3** |

acteristics are essential for applications requiring flexible processing of variable-length video inputs within constrained computational environments.

## D    LIMITATION AND FUTURE WORK

Although CrossLMM has demonstrated promising results in video understanding, an important future direction lies in extending its framework to 2D image and 3D point cloud understanding. This expansion is particularly crucial for handling high-resolution images and large-scale 3D scenes, where capturing fine-grained spatial details while maintaining processing efficiency within LLMs remains challenging.

## E    BROADER IMPACTS

CrossLMM improves the efficiency of large multimodal models processing video, which may allow wider adoption in real-world applications such as smart surveillance, healthcare, and education. However, the enhanced accessibility of such technologies also raises potential ethical concerns, including risks to privacy and misuse for malicious purposes. We encourage responsible development and deployment of these systems, with attention to fairness, privacy protection, and prevention of harmful applications.

Table 6: **Computational efficiency metrics for CrossLMM-2B across varying temporal resolutions**

| Model | Frame Count | CUDA Memory (MB) | FLOPs (TFLOPs) | Prefill Time (ms) |
|---|---|---|---|---|
| CrossLMM-2B | 32 | 715.61 | 10.93 | 264.06 |
| CrossLMM-2B | 64 | 825.21 | 21.77 | 505.50 |
| CrossLMM-2B | 128 | 1046.68 | 43.46 | 909.77 |
| CrossLMM-2B | 256 | 1490.25 | 86.82 | 1698.12 |

