# OpenReview forum: "CrossLMM: Decoupling Long Video Sequences from LMMs via Dual Cross-Attention Mechanisms"
_ICLR.cc/2026/Conference — ICLR 2026 Conference Withdrawn Submission_

### Official Review · Reviewer_GCGM · 2025-10-26

**Soundness:** 2
**Presentation:** 3
**Contribution:** 2
**Rating:** 4
**Confidence:** 4

**Summary:**

This paper explores methods to reduce the number of visual tokens used in large visual (video) language models, aiming to improve efficiency without compromising performance. To achieve this, the authors propose two new modules: a visual-to-visual cross-attention module (V2V) and a text-to-visual cross-attention module (T2V). The V2V module performs hierarchical visual aggregation, moving from coarse- to fine-grained representations while being guided by textual information, enabling more effective selection and refinement of visual features. Meanwhile, the T2V module strengthens interactions between textual input and the original visual tokens, enhancing the model’s multimodal comprehension. Experimental results show that the proposed method attains performance comparable to state-of-the-art approaches while greatly reducing the number of visual tokens, highlighting its potential to improve computational efficiency in large-scale video-language applications.

**Strengths:**

1.	The proposed approach is clearly explained.
2.	It delivers performance on par with existing methods using much fewer visual tokens.
3.	The method’s effectiveness is demonstrated across several benchmark datasets.

**Weaknesses:**

1. The proposed CrossLLM framework offers limited novelty, as the idea of a cross-attention-based language model have been widely explored in previous works. The paper does not clearly explain how CrossLLM provides a significant extension or meaningful differentiation from previous works.

2. The token compression is applied only within individual frames, which may overlook a key characteristic of the video modality— the redundancy of information across frames. It would be interesting to explore how this method could be extended to compress visual tokens across multiple frames.

3.Although CrossLMM effectively compresses tokens, the performance degradation remains considerable, as shown in Table 1.

**Questions:**

1.In the ablation study, the authors examine the effects of V2V insertion frequency and the training strategy for the T2V module. However, I wonder about the necessity of maintaining two distinct modules. Could a unified cross-attention mechanism be employed to jointly update both visual and textual tokens? Including a comparison or discussion of this design choice would provide valuable insights into the potential trade-offs and rationale behind the current approach.

2. In the introduction of the visual-language projector, the authors mention using a single projector along with a differentiable inverse approximation. However, the paper lacks details on this inverse approximation—specifically, how it is implemented or computed.

---

### Official Review · Reviewer_JsVk · 2025-10-29

**Soundness:** 2
**Presentation:** 2
**Contribution:** 2
**Rating:** 4
**Confidence:** 4

**Summary:**

This paper introduces CrossLMM, a framework that uses a dual cross-attention mechanism (visual-to-visual and text-to-visual) to efficiently compress the number of visual tokens for processing long videos in Large Multimodal Models (LMMs), aiming to maintain high performance while significantly reducing computational cost.

**Strengths:**

- The proposed dual cross-attention mechanism is a well-motivated design, as it aims to compress visual information by considering both the global visual context (V2V) and the guiding semantic information from the text (T2V), which could lead to more informed and effective token selection.

- The paper is well-written and easy to follow.

**Weaknesses:**

1. The core contribution of the paper is a dual cross-attention module that incorporates both global visual and textual information when compressing visual tokens. However, the idea of using cross-attention for visual token compression is not novel and has been explored in prior works, such as [1 2 3].

[1] RSTNet: Captioning with Adaptive Attention on Visual and Non-Visual Words

[2] Hybrid-Level Instruction Injection for Video Token Compression in Multi-modal Large Language Models

[3] LLAVA-MINI: EFFICIENT IMAGE AND VIDEO LARGE MULTIMODAL MODELS WITH ONE VISION TOKEN

2. The method section devotes considerable space to describing standard techniques in MLLMs, such as the bilinear pooling operator, which is widely used in models like LLaVA-OneVision, Qwen2.5-VL, and InternVL2.5. These components are not novel contributions of this work.

3. Regarding the experiments:
   • In Table 1, results are shown for CrossLMM 7B under different token counts in the "per frame" setting. Does the model need to be retrained for each different token number?

   • In Table 2(a), the performance gain from using T2V alone is quite limited—only 0.1 on VideoMME and 0.2 on MVBench. However, when T2V is combined with V2V, the improvement on MVBench reaches 1.0. Could the authors provide a detailed explanation for this phenomenon?

**Questions:**

See Weaknesses

---

### Official Review · Reviewer_V5Em · 2025-10-31

**Soundness:** 2
**Presentation:** 2
**Contribution:** 2
**Rating:** 2
**Confidence:** 4

**Summary:**

This paper proposes CrossLMM, a model designed for video understanding tasks that leverages both visual-to-visual and textual-to-visual cross-attention mechanisms to learn alignments between pooled visual tokens and textual tokens. CrossLMM is evaluated on benchmark datasets covering both short-form and long-form video understanding, demonstrating its effectiveness.

**Strengths:**

* The paper is clearly written and easy to follow.
* The idea is intuitive and well-presented.
* Enhancing the alignment between visual and textual modalities after token reduction is a promising direction worthy of further exploration.

**Weaknesses:**

* While the use of both visual-to-visual and textual-to-visual cross-attention is intuitive and known to improve performance, it is not novel. Also the primary efficiency gain of CrossLMM appears to stem from token pooling, which drastically reduces the number of visual tokens, not from the added attention modules.
* Given that the core architectural idea is well established, a more rigorous experimental evaluation is expected. Yet, I believe the comparison between CrossLMM and the baselines is unfair: CrossLMM employs significantly stronger backbones (Qwen2.5 and SigLIP2), while many baselines rely on older or weaker models. As a result, it is difficult to attribute performance gains solely to the proposed architecture rather than the superior pre-trained components.
* The efficiency comparison is conducted primarily against LLaVA-OV, which not only uses weaker backbones (Qwen2 and SigLIP) but also lacks any token reduction mechanism. This makes it an unsuitable baseline for evaluating efficiency claims. I think it's hard to draw clear conclusions from this set of experiments.
* If efficiency is a central selling point, the paper should include comparisons with other methods that also employ token reduction strategies, under a controlled and fair experimental setup (e.g., using the same backbones and training protocols).
* As both pre-training and SFT are applied, have the authors considered starting with the base version of Qwen rather than the instruct version?

**Questions:**

Please refer to the weaknesses.

---

### Official Review · Reviewer_aQ8W · 2025-11-02

**Soundness:** 3
**Presentation:** 3
**Contribution:** 3
**Rating:** 4
**Confidence:** 3

**Summary:**

The paper proposes a novel scheme for visual token compression for VLM. Instead of compressing the tokens completely before using them as input to the LLM, the paper proposes a scheme where over the different layers of the LLM, cross attention back to the original visual tokens is used to update both the visual tokens and the textual tokens. This allows for a 'language-aware' compression, leading to superior results compared to earlier works.

**Strengths:**

The proposed dual attention mechanism makes sense, differs from earlier attempts that try to compress completely before the first LLM layer, and gives good results in practice. I've always wondered why the visual tokens were treated the exact same way as the textual tokens. Now that seems corrected.

The method has been tested on diverse video benchmarks, including videos of various length.

**Weaknesses:**

1. The math doesn't add up in the section on 'initial token merge', where the use of bilinear pooling is said to reduce the dimensionality by a factor of 9, consistent with a 3x3 local patch aggregation, but then it's said this brings them from N=729 to N=9.  I thought this was a typo, but later in the experiments the authors keep mentioning sizes of 1 / 9 / 16, so now I'm confused and don't know what the authors are actually doing in this phase.

2. One would expect that a more compact handling of the visual tokens would ultimately lead to significantly better results when working with long videos, allowing more frames to be considere, compared to the baseline scheme that can encode only a small fraction of the frames. However, this is not obvious from the obtained results.

**Questions:**

1. Can you better explain the initial token merging (see Weakness 1 above) ?
2. The text suggests that the same gamma is used for both V2V and T2V. Is that correct ? Seems a weird choice.
3. Reducing the memory consumption and computational overhead is nice, but even nicer would be to show that, for a similar memory and compute budget, when dealing with long videos, your method could give significantly better results. Can you show this ? If not, do you know why ?

---

### Note · Authors · 2025-11-14

I have read and agree with the venue's withdrawal policy on behalf of myself and my co-authors.